# Plasticity of growth laws tunes resource allocation strategies in bacteria

**Avik Mukherjee**◉⊗, **Yu-Fang Chang**⊗, **Yanqing Huang**◉, **Nina Catherine Benites,**
**Leander Ammar**◉, **Jade Ealy, Mark Polk, Markus Basan**◉*

Department of Systems Biology, Harvard Medical School, Boston, Massachusetts, United States of America

⊗ These authors contributed equally to this work.
* markus@hms.harvard.edu

**Data Availability Statement:** Raw data available at: https://data.mendeley.com/datasets/tz2jyv6y3d/1.

**Funding:** This project was supported by MIRA grant (5R35GM137895) and an HMS Junior Faculty Armenise grant to MB. N.C.B was

## Abstract

Bacteria like *E. coli* grow at vastly different rates on different substrates, however, the precise reason for this variability is poorly understood. Different growth rates have been attributed to 'nutrient quality', a key parameter in bacterial growth laws. However, it remains unclear to what extent nutrient quality is rooted in fundamental biochemical constraints like the energy content of nutrients, the protein cost required for their uptake and catabolism, or the capacity of the plasma membrane for nutrient transporters. Here, we show that while nutrient quality is indeed reflected in protein investment in substrate-specific transporters and enzymes, this is not a fundamental limitation on growth rate, at least for certain 'poor' substrates. We show that it is possible to turn mannose, one of the 'poorest' substrates of *E. coli,* into one of the 'best' substrates by reengineering chromosomal promoters of the mannose transporter and metabolic enzymes required for mannose degradation. This result falls in line with previous observations of more subtle growth rate improvement for many other carbon sources. However, we show that this faster growth rate comes at the cost of diverse cellular capabilities, reflected in longer lag phases, worse starvation survival and lower motility. We show that addition of cAMP to the medium can rescue these phenotypes but imposes a corresponding growth cost. Based on these data, we propose that nutrient quality is largely a self-determined, plastic property that can be modulated by the fraction of proteomic resources devoted to a specific substrate in the much larger proteome sector of catabolically activated genes. Rather than a fundamental biochemical limitation, nutrient quality reflects resource allocation decisions that are shaped by evolution in specific ecological niches and can be quickly adapted if necessary.

## Author summary

Bacteria grow at very different rates on different substrates. Therefore, the substrates themselves are often denoted as 'rich' substrates versus 'poor' substrates. However, it remains unclear what makes a nutrient a 'rich' or a 'poor' substrate. A host of different explanations have been suggested, such as the energy content and chemical composition of the nutrient, the amount of protein required for efficient catabolism of the substrate, or

supported by following grants, National Science Foundation Graduate Research Fellowship Program (DGE 2140743) and Systems, Synthetic, and Quantitative Biology Training grant award (T32GM135014). Any opinions, findings, and conclusions or recommendations expressed in this material are those of the author(s) and do not necessarily reflect the views of the National Science Foundation. The funders had no role in study design, data collection and analysis, decision to publish, or preparation of the manuscript.

**Competing interests:** The authors have declared that no competing interests exist.

limitations of 'space' in the plasma membrane for fitting transporters of the substrate. All of these hypotheses are actively debated today. Here, we show instead that, at least for certain substrates, nutrient quality can be a plastic property that can be dialed by evolution. Different nutrients enable bacteria to grow, but they also serve as a major signal that allows microbes to infer information about their environment. We propose that nutrient quality encoded in a combination of regulatory architecture and enzymatic properties serves both as a map of the safety and reliability of the environment and as a regulatory mechanism that implements proteome allocation decisions.

## Introduction

Different growth rates of bacteria on different substrates are often attributed to the 'quality' of nutrients, which is closely related to carbon hierarchy in catabolite repression [1,2]. 'Nutrient quality' is the key parameter in bacterial growth laws [3–5], originally formulated by Scott et al. [3,4], which elegantly link growth rates and proteome allocation. However, the precise meaning of this growth-rate-determining parameter remains unclear.

As illustrated in Fig 1A, growth laws originate from the balance of two fluxes: First, protein synthesis flux from translating ribosomes generates the increase in total protein proportional to the growth rate $\lambda$:

$$\lambda \sim \kappa_t(\phi_R - \phi_0), \tag{1}$$

where $\phi_R$ is the ribosomal fraction of the proteome. Note that $\phi_R$ implicitly includes the rest of protein biosynthesis pathways, including amino acid biosynthesis, whose protein cost theoretically scales identical to ribosomes. $\phi_0$ is a constant, and $\kappa_t$ is a parameter denoting the

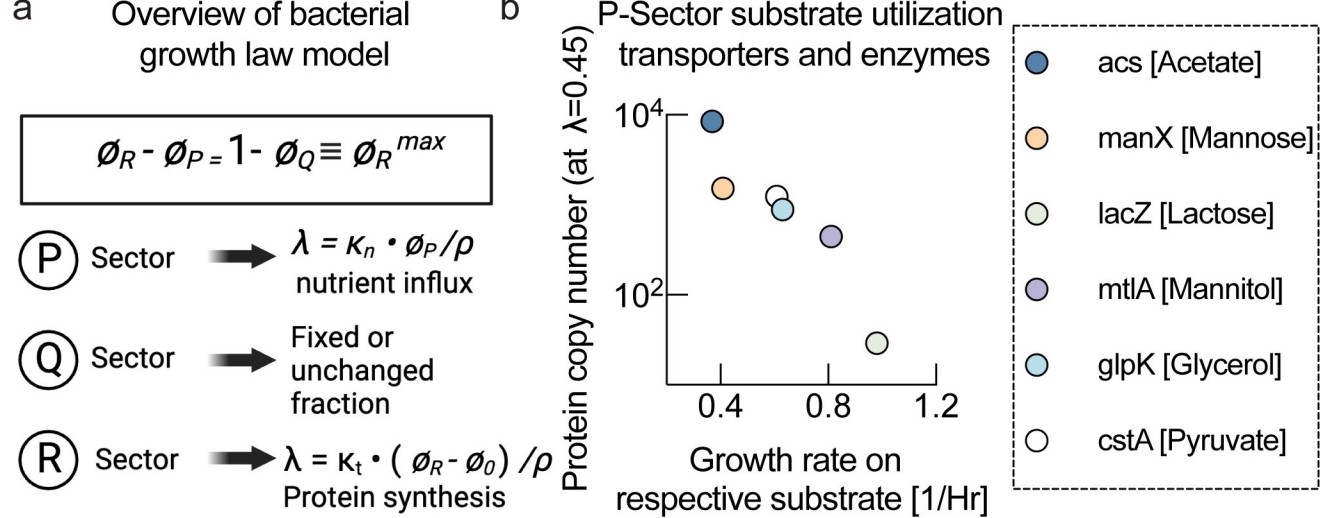

**Fig 1. Growth laws and protein investment. a,** Overview of bacterial growth law model. Growth rate is determined by the balance of nutrient influx catalyzed by the P-sector and nutrient consumption by protein biosynthesis catalyzed by the R-sector. The Q-sector is a constant growth-rate-independent part of the proteome. Nutrient quality $\kappa_n$ determines how much nutrient flux is achieved for a given P-sector fraction. For higher nutrient quality ($\kappa_n$ large), a smaller P-sector achieves sufficient nutrient flux and frees up proteome for the R-sector for more biosynthesis, resulting in a higher growth. (panel created with Biorender.) **b,** Protein copy number of transporters (or first metabolic enzyme) of different substrates plotted against growth rate achieved on these respective substrates. Protein copy numbers were calculated for the same slow carbon-limited growth conditions by combining proteomics [6] and ribosome profiling data sets [12].

translational capacity, proportional to the speed of ribosomal elongation. Second, biomass building blocks for biomass production rate (proportional to growth rate) are sustained by an equivalent nutrient influx

$$\lambda \sim \kappa_n \phi_P, \tag{2}$$

where $\phi_P$ is called the P-sector, the proteome fraction that includes 'catabolic' enzymes catalyzing this nutrient flux. Here, $\kappa_n$ is the key parameter, denoting the 'nutritional capacity' or the 'nutrient quality' of the substrate. Hence, both nutrient flux and translational flux promote faster growth. However, the proteome sectors $\phi$ are fractions of the total proteome and are therefore constrained. In other words, the sum of these sectors cannot surpass a maximum fraction, denoted by $\phi_R^{max}$, which could theoretically by equal to 100% of the proteome, but is found to be smaller than 100% empirically:

$$\phi_P + \phi_R = \phi_R^{max}. \tag{3}$$

Eqs [1–3] can be combined to give an expression relating the growth rate, the translational capacity, and the 'quality' of the substrate:

$$\lambda = \lambda_{max} \frac{\kappa_n}{\kappa_n + \kappa_t}, \tag{4}$$

where $\lambda_{max}$ resembles a maximum growth rate that is related to the growth-rate invariant fraction of the proteome $\phi_{fixed}$ via $\lambda_{max} = (1-\phi_{fixed})\kappa_t$.

As neither translational capacity $\kappa_t$, nor maximum growth rate $\lambda_{max}$ depend on the carbon substrate, according to Eq [4], the key growth rate-determining parameter in the growth laws is the nutrient quality $\kappa_n$. As illustrated in Fig 2B, this parameter determines the size of the P-sector at maximum rate. Growth on a 'good' substrate (large $\kappa_n$) results in a small P-sector, freeing up more proteomic resources for ribosomes (R-sector) and thereby allowing faster growth (Fig 2B, right side). Conversely, a 'poor' substrate (small $\kappa_n$) results in a larger P-sector, constraining the size of the ribosomal R-sector and therefore resulting in slower growth (Fig 2B, left side).

Growth laws make a set of elegant predictions that have been quantitatively validated experimentally [3,4] and form the basis of many more detailed models [1,6,7]. However, the basic question how the growth rate determining parameter nutrient quality $\kappa_n$ should be interpreted remains. Nutrient quality could be an intrinsic carbon-specific property, such as proteome cost of the catabolic pathways, required for metabolizing the substrate and therefore determined by fundamental enzymatic properties like the maximum catalytic rate $V_{max}$ of these metabolic enzymes and transporters. In this case, growth on a 'poor' substrate leads to an obligatory increase in the P-sector $\phi_P$ that limits growth rate. The other possibility is that nutrient quality $\kappa_n$ is a 'plastic' parameter, largely determined by the relative expression levels of carbon-specific transporters and enzymes. In this case, bacteria are programmed by evolution to grow slowly on certain substrates. Different species of bacteria and even different strains could set 'nutrient quality' at different levels and grow at different rates. It should then be possible to change growth rates by simply changing the expression level of carbon-specific transporters and metabolic enzymes. However, despite the simplicity of this idea, it is challenging to test experimentally because metabolic pathways of individual substrates typically involve many enzymes and transporters that need to be expressed in concert, as well as additional layers of transcriptional and allosteric regulation that affect expression levels and fluxes.

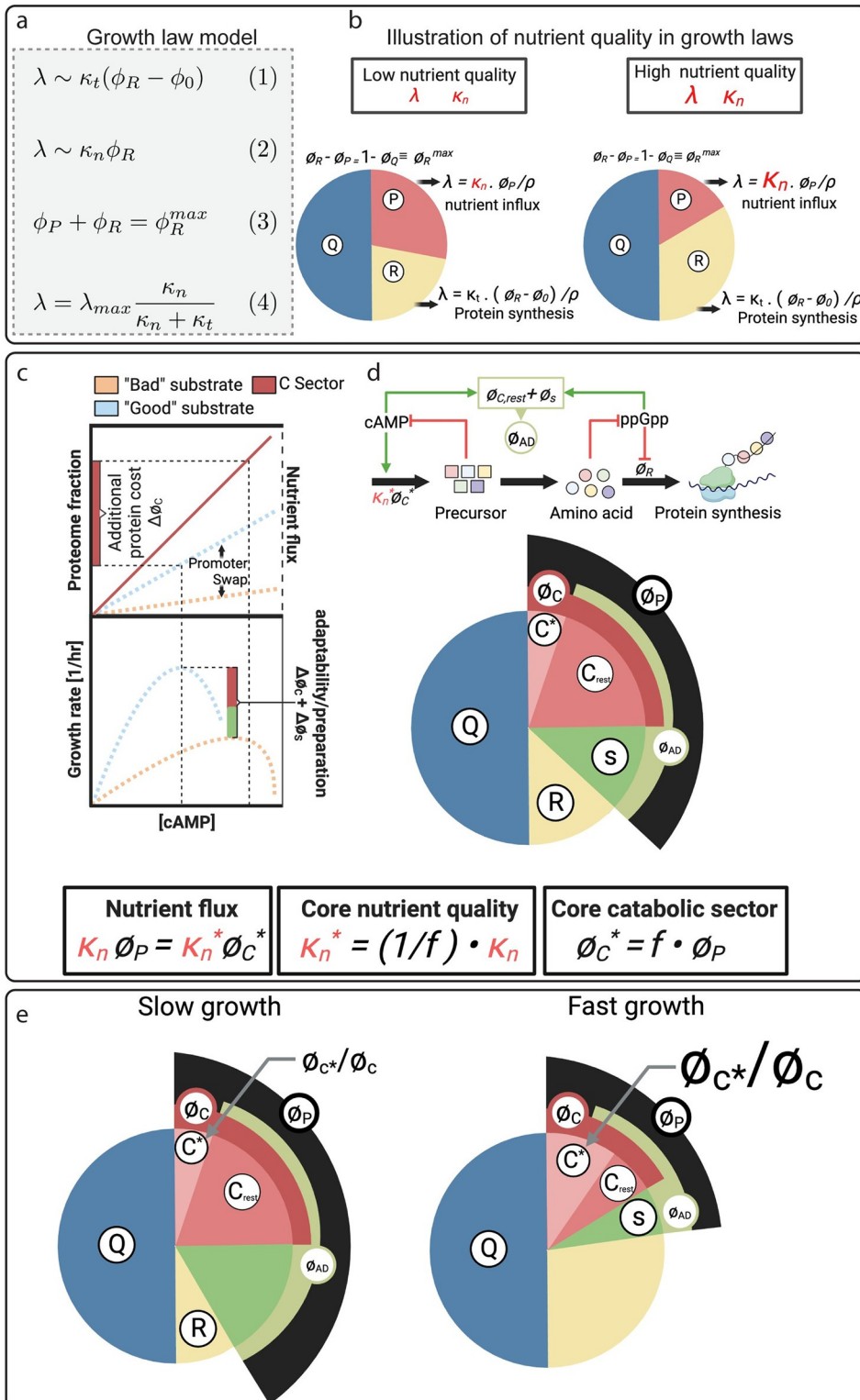

**Fig 2. Model summary. a,** Summary of the growth law model and parameters. $\phi_R$ is the ribosomal fraction of the proteome. $\phi_0$ is a constant, and $\kappa_t$ is a parameter denoting the translational capacity. $\phi_P$ is called the P-sector, the proteome fraction that includes 'catabolic' enzymes catalyzing this nutrient flux. $\kappa_n$ is the 'nutritional capacity' or 'nutrient quality'. The sum of proteome sectors cannot surpass a maximum fraction, denoted by $\phi_R^{max}$. $\lambda_{max}$ resembles a maximum growth rate that is related to the growth-rate invariant fraction of the proteome $\phi_{fixed}$ via $\lambda_{max} = (1-\phi_{fixed})\kappa_t$.

**b**, Illustration how different nutrient quality results in different growth rates. For a low nutrient quality (left), a high expression level of the P-sector is required to achieve sufficient nutrient flux. This leaves fewer resources available for the ribosomal sector $\phi_R$ and overall results in slower growth. Conversely, for high nutrient quality $\kappa_n$ (right), a higher nutrient flux is catalyzed by a smaller P-sector, freeing up proteomic resources for higher expression of the ribosomal sector $\phi_R$ and resulting in faster growth. **c,** Illustration of nutrient flux and growth rate as a function of cAMP-mediated C-sector expression (left). The C-sector is one of the major components of the P-sector in the growth theory, as illustrated in the pie chart (right). Another large part being the ppGpp-activated protein sector that we denote as the S-sector, where S stands for stress. Most transporters and substrate-specific metabolic genes are part of the C-sector and increasing the C-sector increases nutrient flux (dashed lines, left top panel). Higher nutrient flux leads to an increase in growth rate, but only up to an optimum level, at which flux for biosynthesis balances nutrient flux (left bottom panel). At even higher C-sector expression, growth rate drops because there are insufficient proteomic resources for the R-sector to process nutrient flux generated by the disproportionate C-sector. Nutrient quality $\kappa_n$ is determined by how much nutrient flux is achieved per C-sector induction. A "good" substrate results in a steep increase in nutrient flux (blue dashed line, left top panel), whereas a "poor" substrate results in a much flatter induction of nutrient flux (orange dashed line, left bottom panel). Therefore, "poor" substrates result in higher cAMP levels but slower growth rates than "good" substrates (left bottom panel). The steepness of nutrient flux induction, defining nutrient quality, is determined by the catalytic rates of substrate-specific enzymes, but also by the expression level of substrate-specific transporters and enzymes. We denote the core proteome fraction of substrate-specific transporters and enzymes by $C^*$-sector, which is a fraction f in the much larger P-sector, $f = \phi_{C^*}/\phi_P$. We then denote the core nutrient quality based on fundamental biochemical enzymatic properties by $\kappa_n^*$. The effective nutrient quality that emerges in the growth laws can be modulated by changing the expression fraction f, $\kappa_n = f\ \kappa_n^*$. We hypothesize that because the core enzyme proteome fraction $\phi_{C^*}$ is a small fraction of the P-sector $\phi_P$, even for the costliest substrates in terms of protein cost, nutrient quality can be dialed up or down in response to ecological needs by changing the expression fraction f. We denote the P-sector fraction that is not part of substrate-specific metabolism as the "adaptability" sector $\phi_{AD}$: $\phi_{AD} = \phi_P - \phi_{C^*}$. Components of this sector are not important for supporting growth in the current growth conditions, but instead constitute a preparatory response. **d,** Illustration how the core catabolic fraction determines nutrient quality and growth rate. Within the co-regulated C-sector, low expression and a weak induction of substrate-specific enzymes will result in a lower effective nutrient quality and a higher expression level of the adaptability sector $\phi_{AD}$ (left). Conversely, a large core catabolic fraction results in fast growth and relatively low expression of the adaptability sector $\phi_{AD}$ (right). By dialing the core catabolic fraction, bacteria can convert information about their environment conveyed by the nutrient present, into resource allocation decisions determining their adaptability and preparedness for changing environments or the onset of stress. (illustrations created with Biorender).

## Materials and methods

### Preparation of bacterial growth media

For the experiments mentioned in this study, we used single carbon and single nitrogen-source containing minimal media denoted N+C+ minimal medium. In this minimal medium, 20mM $NH_4Cl$ was maintained as single nitrogen source, and 20 mM glucose or 20mM mannose were used as carbon sources as required.

To make the minimal media, a 4X N-C- salt stock-solution was first prepared, and the recipe to prepare 1L 4X N-C- solution is shown below.

| Species | Grams (g) | Formula Weight (Da) | Molarity (mM) |
|---|---|---|---|
| $K_2SO_4$ | 4 | 174.26 | 23.0 |
| $K_2HPO_4$ | 54 | 174.18 | 310.0 |
| $KH_2PO_4$ | 18.8 | 136.09 | 138.1 |
| $MgSO_4$ | 0.192 | 120.37 | 1.6 |
| NaCl | 10 | 58.44 | 171.1 |

To prepare 1X N+C+ minimal media, 4X N-C- salt solution stock was diluted to 1X concentration in final media formulation and supplemented with 20 mM $NH_4Cl$ (final concentration), and the carbon source (20mM glucose or 20 mM mannose) was added separately. After preparation, each medium was filtered using disposable vacuum filtration system (Corning,

PES filter with 0.22 pore size). Carbon sources were added from 1M stock solution, prepared from D (+) Mannose (Sigma Aldrich, M6020-25G, Lot #BCBV4824), and glucose (Sigma, G5146).

## YCE119 construction

We replaced the chromosomal promoter driving the D-mannose PTS permease manXYZ with the promoter of the glucose PTS transporter PptsG, and deleted the mlc gene in the same strain.

For strain construction, we used NCM3722 as background.

Strain genotype: NCM3722 FRT-Kan-FRT:rrnBT:PptsG-m5'UTR-manX, Δmlc, FRT-FRT: Ptet-manA.

The mlc gene was knocked out with the mlcΔ p1 lysate [8] generated from Keio collection [9]; the kanamycin cassette was then flipped out with the one-step inactivation method [10]. For integrating the Tet promotor to manA gene chromosomally, the pkD13 plasmid was used to amplify the Ptet promotor with primers that also contain manA target region for replacing manA's own chromosomal promotor. The PCR product was sequenced before chromosomally integrated into the target strain using pKD46 recombineering method, and the kanamycin cassette was flipped out [10]. The promotor of the glucose PTS transporter PptsG was first cloned into the pKD13 plasmid using Gibson Assembly. The Gibson Assembly product was then used as the PCR template with primers that span from ptsG promotor to manX gene for PCR amplification and included homologous regions for integration. The PCR product was sequenced before being chromosomally integrated into the target strain using pKD46 recombineering method [10] and selected with kanamycin.

## Bacterial culture and growth rate measurement

A single isolated colony of WT and YCE119 was picked up from respective LB plates and inoculated in glass tubes containing 5 ml LB and grown for 2–3 Hours. For biological replicates, multiple tubes were inoculated, each from different single colonies. Inoculated LB tubes were incubated in a shaking air incubator (infros HT), set at 37˚C with 200 rpm orbital shaking. After 2–3 hours of growth in LB, 1% inoculum was transferred to glass tubes containing fresh minimal media (N+C+glucose or N+C+mannose) and incubated in the shaker incubator overnight. In this way, the same colony can be propagated in two different types of media for downstream experiments. Next morning, from the overnight in minimal media, 1% inoculum was taken out and transferred to glass tubes containing 5 ml fresh minimal media (glucose or mannose respectively). Freshly inoculated tubes were placed in shaker water bath set at 37˚C with 200 rpm orbital shaking.

For experiments with cyclic AMP, we added cAMP directly to the glass tubes containing fresh minimal media from a 25 mM cAMP stock solution (cyclic AMP sodium salt, Sigma, A6885), and then the tube was inoculated with desired bacterial strain.

For growth rate measurements, for each biological replicate, samples were taken for at least 4 time points during steady-state growth, and optical density ($OD_{600}$) was measured using a spectrophotometer (Genesys 30 visible spectrophotometer, Thermo Scientific). Doubling time and growth rates were determined by plotting the measured OD as a function of elapsed time in a semi-log scale.

## Range extension assay

Range extension assay was done following the protocol modified from Cremer et. al. [11]. To prepare soft agar plates, we mixed 0.25% agar in 1X N+C- minimal media and autoclaved the

agar-containing media. Carbon sources were not added at the time of autoclaving to avoid any denaturation and caramelization of the sugars. 1M glucose and 1M mannose solution were prepared separately and sterilized by passing through a 0.22μm filter. Autoclaved soft agar and carbon stock solutions (glucose and mannose) were taken inside a sterile biosafety cabinet, and we added glucose and mannose separately to the soft agar to obtain a final glucose or mannose concentration of 20 mM. After addition of respective carbon sources, 15 ml soft agar was poured on 100mm sterile plastic petri plates and kept inside a laminar airflow biosafety cabinet to maintain sterility. Note, unlike Cremer et. al. no attractant was added to soft agar plates. For cAMP containing soft agar plates, cAMP was added to still warm soft agar (after addition of respective carbon sources) from a 25mM stock solution (cyclic AMP sodium salt, sigma), as mentioned before, to obtain a final concentration of 3.5 mM. 7 ml warm soft agar was poured on 60 mm plates inside a laminar airflow biosafety cabinet.

After pouring into plates, soft agar was allowed to cool off, and when solidified, plate lids were placed, and plates were taken out of the biosafety cabinet. Due to the soft consistency of agar, these plates cannot be inverted.

For range extension assay, WT (NCM3722) and YCE119 cultures were grown in minimal media with 20mM glucose, 20mM mannose, and 20mM mannose with 3.5 mM cAMP, following the protocol mentioned above. When respective cultures were still in the exponential growth phase and reaching ~0.4–0.5 OD, 2 μl of culture was taken out of each exponentially growing tube and gently placed on the pre-warmed (to 37˚C) soft agar plates as a droplet. On each plate, 4 droplets were placed equidistant from one another. Plates were carefully placed in a 37˚C air incubator and were not inverted. After overnight incubation, plates were taken out and imaged in a custom-built plate imaging set-up. After imaging, plates were placed back in the incubator and re-imaged on day 3. We kept the incubator humidified by keeping an open water pan to ensure plates did not dry out during long incubation periods. Water droplets inside the lid were removed by soaking with sterile absorbent paper during the incubation period.

For colony size measurement, plate images were analyzed in FIJI. Each colony image was binarized by applying a threshold, and the colony area was calculated from the binary image.

## Starvation assay

WT (NCM3722) cells were grown exponentially in minimal media containing 20mM glucose or 20 mM mannose, and YCE119 cells were grown in minimal media containing 20 mM mannose or 20 mM mannose and 3.5 mM cAMP. When cells were still in the exponential phase (0.2–0.4 OD), an aliquot from the exponentially growing culture was taken out for the starvation assay. To keep the cell number roughly similar at the beginning of starvation assay, volume of sample taken was adjusted according to optical density. Cells were centrifuged and pelleted down, and resuspended in carbon free minimal media (N+C-). This step was repeated twice to remove traces of carbon sources. Cells were finally resuspended in 5 ml N+C- starvation media, lacking the carbon source, in glass tubes and placed in a shaking air incubator set at 37˚C at 200 rpm. Immediately after resuspension of the cells in carbon free medium at the beginning of the starvation assay, samples were taken out of each tube and diluted 100,000-fold in carbon free medium. 100μl of the diluted samples from each tube were plated on LB-agar plates containing tetrazolium chloride. Metabolically active live cells give red-colored colonies on these plates. After plating, plates were incubated in 37˚C air incubator overnight and imaged next morning using a custom-built plate imaging setup. Starving cultures in carbon free media were kept in the shaking incubator for a week, and at the end of experiment, the number of surviving cells were quantified by plating on tetrazolium chloride containing

LB plates. On the final day of the experiment, cells were diluted 20,000-fold in carbon free media and plated. Plates were incubated in a 37˚C incubator overnight and imaged next morning using our custom-built setup. The number of colonies on each plate was counted using Cell Profiler (open-source image processing software) and normalized to colony number of the same condition in the beginning of the experiment, from which survival percentages were calculated.

### Diauxic shift lag time measurement

WT (NCM 3722) and YCE119 strains were grown in LB culture from single isolated colonies for 2–3 hours in a shaking (200 rpm) air incubator set at 37˚C. For biological replicates, multiple isolated colonies (n = 3 for NCM 3722 and n = 3 for YCE119) were inoculated in LB. After growing for a few hours in LB, 1% inoculum was transferred to minimal medium containing 1% (approximately 555μM glucose or mannose). WT strain was inoculated in N+C+555μM glucose medium, and YCE119 was inoculated in N+C+555μM mannose medium. Inoculated tubes were placed in a shaking air incubator (Infors HT) maintained at 37˚C with 200rpm orbital shaking. The next day, 1% of the overnight cultures were freshly inoculated separately in 5ml fresh N+C+ minimal media with 555 μM glucose (WT) or 555 μM mannose (YCE119). From the overnight culture, the YCE119 strain was also inoculated in minimal media with 555uM mannose and supplemented with 3.5 mM cyclic AMP (added from 25mM aqueous stock solution). To measure WT (NCM3722) in mannose, we inoculated the 1% inoculum from the LB cultures (same 3 biological replicate colonies) to 20mM mannose containing minimal medium early in the morning and grown in a shaking air incubator. When these cultures reached exponential phase of growth and ~0.5 OD, 1% inoculum was transferred to minimal medium with 555 μM mannose. Lag phase measurement for WT in mannose was done on a separate plate.

To measure lag phase and growth in acetate after diauxic shift, 5mM (final concentration) sodium acetate were added to each media condition mentioned above and mixed well by vortex mixing. Freshly inoculated tubes were grown for 2–3 hours in an air incubator, and after that 200 μl samples were transferred to a clear bottom 96 well plate with dark wall (Greiner bio-one, 655090). For each biological replicate, multiple wells of the 96 well plate were used. $OD_{600}$ from each well was measured over time in a BioTek Synergy H1 microplate reader, maintained at 37˚C with continuous double-orbital shaking. Background subtracted averages of the $OD_{600}$ from replicate wells for each biological replicate were calculated first, and then the average of all biological replicates with standard deviation was used for plotting. To quantify lag time, each condition was normalized to the point when the exponential growth slowed down for each strain, indicating the depletion of glucose and initiation of diauxic shift. Time of normalization for each strain was denoted as time 0, and normalized fluorescence intensity as a function of time was plotted in a semi-log scale.

### Calculation of P sector allocation and protein cost

To estimate protein copy numbers of the main carbon transporting enzyme (or the first enzyme in the primary carbon degradation pathway) at slow growth rate (0.45), where the P-sector proteins are highly expressed, we combined protein copy numbers determined via ribosome profiling for *E. coli* grown in glucose minimal medium by Li et al. [12] (see S1 Table) with fold-changes between different growth rates determined by $^{15}N$ labeling and mass spectrometry in Hui et al. [6] (see S2 Table) for those proteins that showed up in both datasets. Growth rate in respective substrates (acetate, mannose, lactose, glycerol, and pyruvate) taken

from You et. al. [1], except for growth rate of the wildtype strain on mannitol, which needed to be measured in our lab.

## Results

First, we wanted to determine to what extent nutrient quality of different substrates was reflected in proteome allocation to substrate-specific transporters and enzymes. If nutrient quality were an intrinsic cost of carbon catabolism, poorer substrates would require a larger proteome fraction allocated to them during growth on these substrates. However, there is currently no dataset available that allows comparison of absolute protein abundances of substrate-specific catabolic proteins on many different substrates. To address this question, we wanted to compare protein copy numbers of different P-sector proteins at the same growth rate. Specifically, we were interested in transporters and metabolic enzymes catabolizing different substrates and their expression level at slow growth rates when the P-sctor is fully activated. Therefore, we combined absolute protein copy numbers determined via ribosome profiling for *E. coli* grown in glucose minimal medium by Li et al. [12] with relative fold-changes between different growth rates determined by $^{15}N$ labeling and mass spectrometry in Hui et al. [6]. Strikingly, when we plotted the copy number of carbon transporters (alternatively, the first metabolic enzyme in the degradation pathway) versus the growth rate of *E. coli* on these carbon sources, we discovered a rank-ordered anticorrelation (Fig 1B). Transporters and metabolic enzymes related to poorer substrates were expressed at higher copy numbers than those of better carbon sources.

Although there is limited coverage in the proteomics dataset, and we only analyzed the transporters or first enzymes in the substrate utilization pathways, this finding suggests a relationship between nutrient quality and protein investment. In fact, if nutrient quality were directly caused by differences in the maximum catalytic rate of enzymes and transporters ($V_{max}$), such an anticorrelation between protein cost and growth rate would follow from the growth laws during growth on these substrates because growth on poor substrates would require more enzymes. The observed anticorrelation is still surprising as these substrates were not present in the experimental growth conditions from which protein abundances were determined. Hence, growth rate on different substrates appears to be reflected in expression levels of transporters and metabolic enzymes of these substrates. We found a similar relation to hold when we converted copy numbers to protein cost by considering the size of the individual proteins (S1 Fig).

Nevertheless, we were skeptical if growth rates on poor substrates were directly limited by protein cost or available capacity on the plasma membrane because proteome fractions dedicated to substrate-specific transporters and enzymes are relatively small, and membrane capacity could be increased by changing cell shape. We therefore wanted to test directly whether growth rates of *E. coli* on mannose are directly limited by the cost of producing the necessary proteins for substrate transport and metabolism, or if they are limited in some other way by the available capacity for substrate transport. We decided to focus on the substrate mannose, which is one of the slowest carbon sources and exhibits the second highest apparent protein cost in the set of substrates in Fig 1B. Whereas growth on acetate requires complete rerouting of central metabolism and reversal of flux in glycolysis to a gluconeogenic mode [13], mannose is a glycolytic substrate that requires only the expression of only a handful of dedicated transporters and enzymes. Our strategy was to try to simultaneously alleviate all potential bottlenecks in mannose catabolism to achieve a faster growth rate. To change expression levels while avoiding the burden of plasmid expression, which can affect growth rates, we decided to replace the chromosomal promoter driving the D-mannose phosphotransferase system (PTS)

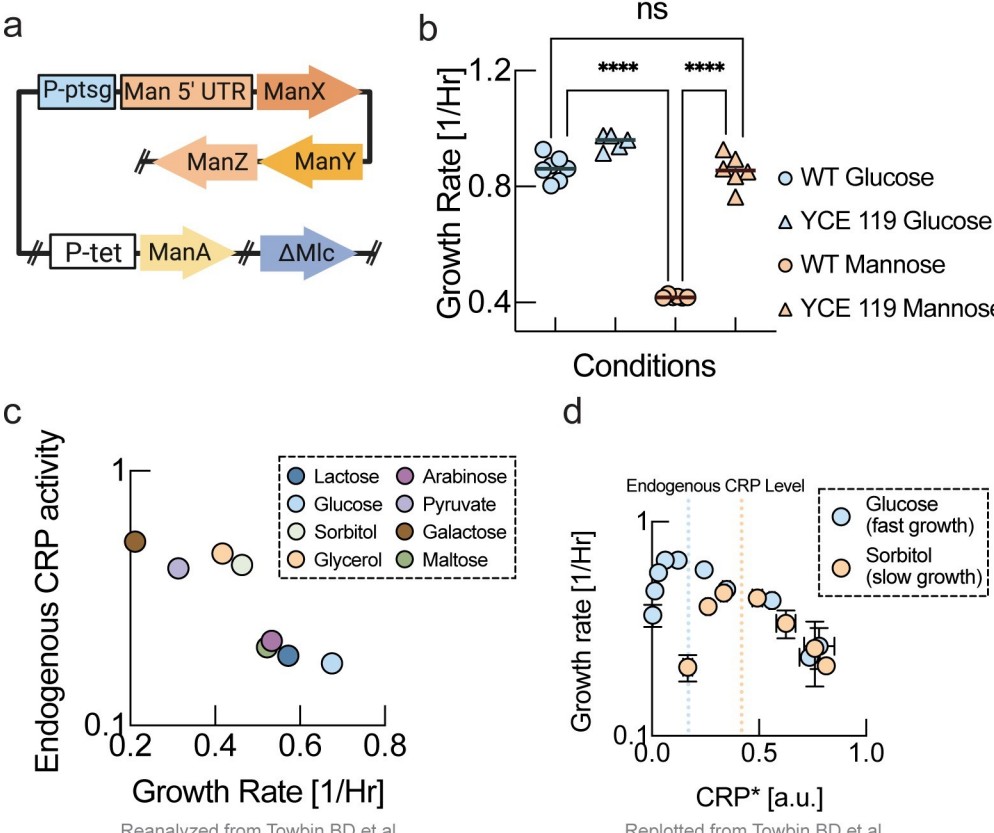

**Fig 3. Plasticity of nutrient quality. a,** We replaced chromosomal promoters of transporters and metabolic enzymes required for mannose metabolism with the promoter of glucose transporter (P-ptsG). To prevent repression due to lack of glucose, we knocked out the glucose-specific transcriptional regulator Mlc. Finally, to ensure that carbon flux from processed mannose enters the glycolysis pathway, we placed mannose-6-phosphate isomerase under a strong constitutive promoter (P-tet). We refer to this strain as the swapped promoter strain (YCE119). (construct map illustration created with Biorender.) **b,** Growth rates of wildtype and swapped promoter strain (YCE119) on glucose and mannose. Mannose is one of the slowest substrates for the wildtype (orange circles), and glucose is often considered the best substrate of E. coli with the fastest growth rates in minimal medium (blue circles). Yet the swapped promoter strain grows on mannose as fast as the wildtype on glucose (orange triangles). Therefore, the genetic modifications in the swapped promoter strain have changed the nutrient quality of mannose into the nutrient quality of glucose. This means that nutrient quality is not limited by fundamental biochemical constraints. Unpaired t-test was performed, and following P-values were obtained. WT Glucose:WT mannose P-value<0.0001. WT Mannose:YCE 119 Mannose P-value<0.0001, and WT Glucose:YCE119 Mannose: non-significant (ns) **c,** Data from Towbin et al. [17], who titrated cAMP levels on different substrates. "Good" substrates have a higher peak growth rate at lower levels of cAMP. **d,** An inverse correlation of endogenous Crp activity on different substrates with growth rate on the respective substrate from Towbin et al. [17]. Poor substrates require higher cAMP levels for maximum growth. A similar relationship is found when plotting CRP activity vs. growth rate for maximum growth rate (S3 Fig).

permease manXYZ with the promoter of the glucose PTS transporter PptsG because glucose is one of the best substrates of *E. coli*. To prevent carbon-specific repression of this promoter due to the lack of glucose, we also deleted the mlc gene. Mlc is a transcription factor that represses the expression of the PTS system dependent on the presence of glucose [14,15]. Finally, to prevent a metabolic bottleneck and to ensure that mannose flux enters glycolysis efficiently, we replaced the chromosomal promoter of the mannose-6-phosphate isomerase gene manA gene with the Tet-promoter. Without expression of TetR, this promoter acts like a strong constitutive promoter [16]. The resulting genetic systems in strain denoted YCE119 is illustrated in Fig 3A and will be referred to as the 'swapped promoter strain'.

Next, we measured growth rates of swapped promoter strain as compared to the wildtype strain in batch culture. If growth rate were truly limited by the protein cost of mannose-specific metabolism or the uptake rate that could be accomplished due to limited capacity of the plasma membrane, overexpression of mannose transporters and metabolic enzymes in the swapped promoter strain should have no positive effect on growth rate. In fact, according to the growth laws, overexpressing too much of the mannose-specific pathway should result in decreased growth rates by resulting in a misbalance of the P-sector and R-sector. Instead, we found that the swapped promoter strain grew on mannose minimal medium at the same rate as the wildtype strain on glucose minimal medium (Fig 3B). While we refer to this strain as the swapped promoter strain for simplicity due to the exchanged promoter of the manXYZ operon, in this strain, the promoter of manA was also exchanged by a strong constitutive promoter (Ptet), and the gene encoding the transcription factor Mlc was knocked out. The effect of subsets of these modifications on growth rate on mannose is shown in S2 Fig. Notably, none of the individual modifications were sufficient to confer the full growth rate advantage. We also found that the strong constitutive promoter (Ptet) driving the manXYZ operon conferred the same growth rate advantage for growth on mannose as the PptsG promoter. These phenotypes vividly demonstrate that mannose is not an inherently 'poor' substrate and that the slow wildtype growth rate on mannose is not the result of any fundamental constraint or limitation. Nutrient quality, at least for the case of mannose, appears to be a plastic property, rather than a fundamental biochemical or enzymatic limitation. Indeed, this result is consistent with previous observations about suboptimal growth rates of *E. coli* on glycerol and fructose [17,18], as well as reports about suboptimal growth rates due to endogenous levels of cAMP [17]. It is also consistent with the observation that substrate preferences can be completely reversed in other bacterial species like *Pseudomonas aeruginosa*, which grow faster on supposedly 'poor' gluconeogenic substrates like acetate than on supposedly 'rich' glycolytic substrates [13], despite the high degree of conservation of central metabolism.

How should nutrient quality on different substrates then be interpreted? We propose that different substrates convey information about the safety and reliability of the current growth environment in the context of the ecological niche via their 'nutrient quality'. Many carbon-specific transporters, metabolic enzymes, and other proteins that are unrelated to the current growth conditions are co-regulated in the cAMP-activated C-sector [1] that is a sub-sector of the P-sector (Fig 2C, red sector). In fact, it is long known that E. coli progressively activates more genes as the 'quality' of carbon source declines, including uptake systems for alternative carbon sources [19]. Moreover, the P-sector includes other preparatory pathways for the response to stress and adverse conditions that are activated by ppGpp, and that we denote the S(tress)-sector (Fig 2C, green sector). As illustrated in Fig 2C and 2D, although being co-regulated, the core C-sector constituting the substrate-specific transporters and enzymes enabling growth on the current growth conditions (denoted the C\*-sector) may constitute only a small fraction of the overall C-sector. According to the growth laws, the entire P-sector is upregulated up to a level, where nutrient flux catalyzed by the core C\*-sector balances the flux of biomass precursors consumed by ribosomes for protein translation via the R-sector. Both too low and too high levels of induction of the P-sector result in slower growth rates as proteome allocation is unbalanced (Fig 2C, left panels), which has been vividly demonstrated experimentally by Towbin et al. who titrated cAMP concentrations [17] (Fig 3C). The cAMP level where the maximum growth rate is achieved depends on the nutrient quality, as can be seen by replotting the maximum growth rate as a function of the corresponding cAMP level from the data by Towbin et al. [17] (Figs 2D and S3).

'Poor' substrates, as compared to 'rich' substrates, require a higher induction of the P-sector to achieve the same nutrient flux and therefore have higher levels of cAMP (Fig 2C, left panel).

However, while this flux depends on the catalytic rates of the substrate-specific enzymes and transporters represented by the core nutrient quality $\kappa_n^*$, their total expression fraction $f$ within the much larger P-sector is just as important and results in the growth-rate determining nutrient quality given by $\kappa_n = f\kappa_n^*$, with $f \leq 1$. Hence, most of the growth-rate-limiting protein cost comes from P-sector components unrelated to current growth conditions (Fig 2D). This is consistent with data from proteomics [6], as well as genome scale metabolic models [20]. In this case, by tuning the expression level of substrate-specific transporters and pathways within the P-sector, nutrient quality of a specific substrate can be freely tuned. This effect is vividly illustrated by the dramatic increase in growth rate on mannose in the swapped promoter strain (YCE 119).

So why would bacteria want to decrease nutrient quality of a given substrate by reducing expression levels of transporters and metabolic enzymes within the P-sector? As illustrated in Fig 4A, lower nutrient quality results in higher levels of the adaptability sector (Fig 2C and 2D). The adaptability sector may not only include many alternative transporters and metabolic enzymes for substrates that are not present in the current medium, but also proteins related to starvation survival, and motility etc. According to this interpretation, 'nutrient quality' reflects a resource allocation decision of fast growth versus preparation for changing growth conditions that is consistent with trade-offs between growth rate and adaptability that have recently been characterized [21,22].

Indeed, using the dataset by Hui et al. [6], we found that with decreasing nutrient quality abundances of proteins shown to be involved in minimizing lag times [22], as well as proteins shown to help starvation survival [23] strongly increased as shown in Figs 4B and 3C respectively. These changes in protein abundances reflect a larger allocation to the adaptability sector $\phi_{AD}$ (Fig 4A). We wanted to quantify if these changes in protein abundances were also reflected in phenotypes directly, and specifically, if the swapped promoter strain exhibited a loss of capabilities as compared to the wildtype strain growing on mannose, reflecting the cost of faster growth. We first tested the ability to adapt to growth on different substrates as quantified by lag time. One of the longest, but also most frequently encountered lag times comes from the shift from glycolytic substrates to the primary fermentation product acetate [21,22]. The wildtype growing on glucose has a much longer lag time in a diauxic shift to acetate than when growing on mannose (Fig 4C, blue circles vs. orange circles). However, as predicted, the swapped promoter strain growing on mannose lost this ability to quickly shift to acetate and exhibited a long lag phase (Fig 4D, orange triangles), comparable to lag time of the wildtype on glucose.

According to the model outlined in Fig 2, it should be possible to reduce lag times by artificially upregulating the C-sector, that is part of the larger adaptability sector, by increasing the abundances of lag related proteins (Fig 4B). Therefore, we measured lag time in a diauxic shift to acetate of the swapped promoter strain with the addition of cAMP to the growth medium, which activates the C-sector. Indeed, the addition of cAMP to the mannose medium resulted in substantially reduced lag of the swapped promoter strain (Fig 4D, orange inverted triangles). However, this improvement in lag times came at the cost of a slower growth rate in the presence of cAMP (see S4 Fig).

Next, we tested the ability of the swapped promoter strain and the wildtype to survive periods of starvation. Bacteria spend a large fraction of their time coping with periods of starvation and therefore adapting to and surviving starvation is likely a key contributor to bacterial fitness in natural environments. After growing bacteria on different substrates, we washed them and resuspended them in carbon-free medium. We then quantified survival by plating and counting colony-forming units at different timepoints during starvation. As anticipated from

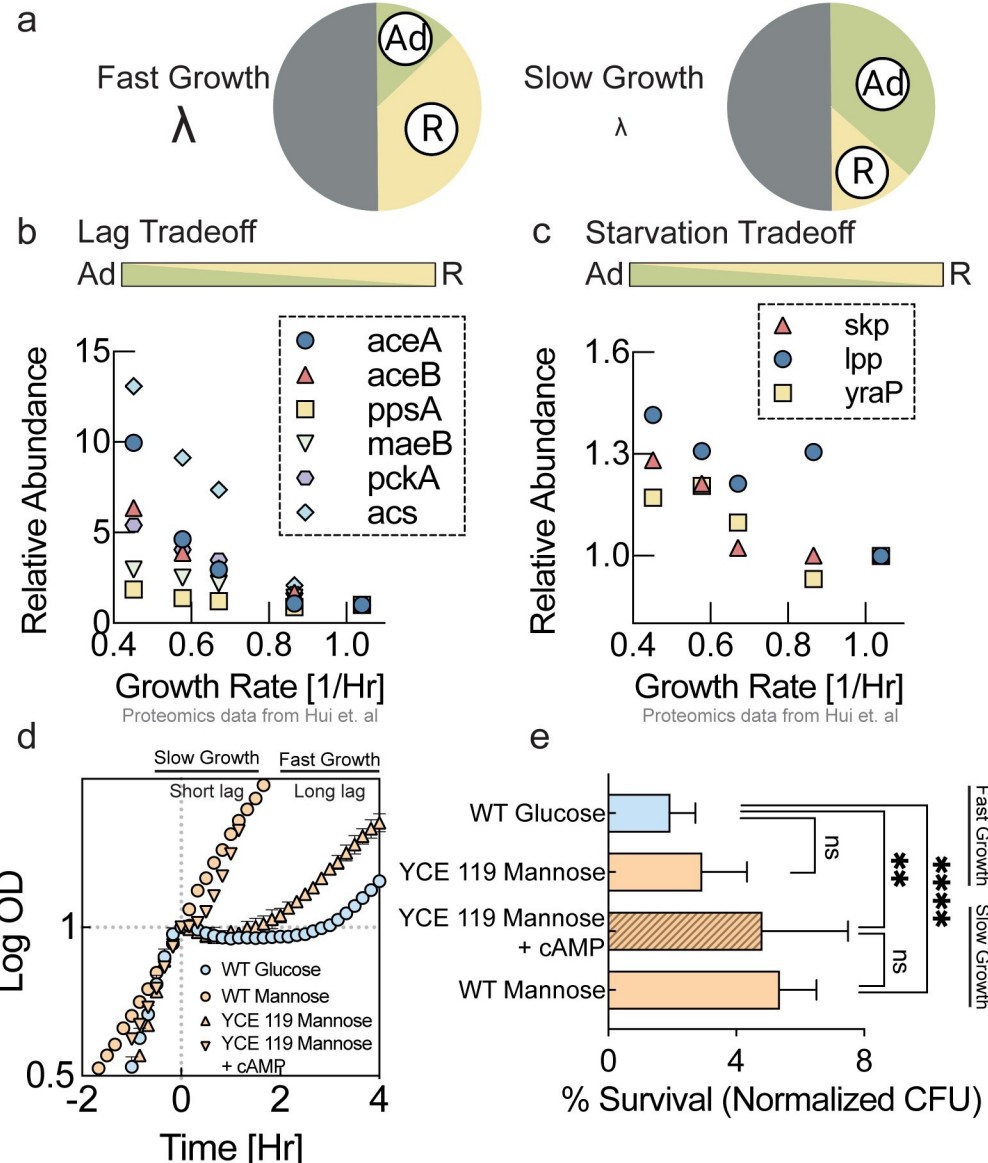

**Fig 4. Lag and starvation cost of higher nutrient quality. a,** Illustration of resource allocation for high nutrient quality (left) and low nutrient quality (right). Low nutrient quality reflects high expression levels of the adaptability sector $\phi_{Ad}$. (pie chart illustrations created with Biorender.) **b,** Relative quantification of proteins that have been shown to be related to lag phase in Basan et al. [22]. Proteomics data from Hui et al [6]. (color bar illustration created with Biorender.) **c,** Relative quantification of proteins that have been shown to be related to improved starvation survival by Schink et al. [23]. Proteomics data from Hui et al [6]. **d,** Diauxic shifts from glucose or mannose to acetate. The wildtype strain exhibits a lag phase of several hours from glucose to acetate (blue circles), but almost no lag phase from mannose to acetate (orange circles). The swapped promoter strain (YCE 119) grows at a similar growth rate to glucose on mannose (Fig 3B) but has lost the ability to quickly switch to acetate. Upregulation of the C-sector by addition of cAMP (3.5mM) to the growth medium results in a much shorter lag time, however, only at the cost of slower growth (S4 Fig), as predicted by the model (Fig 2). Three biological replicates were used for all conditions. Mean values of 3 biological replicates is plotted, and error bars represent standard deviation. **e,** Starvation survival after 7 days of starvation relative to initial CFUs. The wildtype survived carbon starvation much better after growing on mannose (bottom bar) than on glucose (top bar). The faster-growing swapped promoter strain (YCE 119) lost this improved survival on mannose (2nd bar from top). With addition of cAMP (3.5mM) to the mannose growth medium but not to the starvation medium, the swapped promoter strain exhibited improved starvation survival (3rd bard from top) at the cost of a slower growth rate (S4 Fig). Three biological replicates were used for YCE119 grown in N+C+Mannose, WT grown in N+C+Glucose, and 4 biological replicated used for YCE 119 grown in N+C+Mannose with 3.5mM cAMP and WT grown in N+C+Mannose. Every biological replicate was plated in triplicate. Mean survival frequency plotted, and error bars represent standard deviation. Unpaired t-test performed, and following P-values obtained: WT

Glucose-YCE119 Mannose: non-significant, WT Glucose-YCE119 Mannose+3.5mM cAMP:0.0054, WT Glucose-WT mannose<0.0001.

changes in the abundance of starvation-related proteins (Fig 4C), after growing more slowly on mannose minimal medium, the wildtype strain survived substantially better than after faster growth on glucose (Fig 4E, 1st bar vs. 4th bar). However, consistent with our expectation, this improved starvation survival ability after growth on mannose was completely lost in the faster-growing swapped promoter strain (Fig 4E, 2nd bar from the top). According to the model outlined in Fig 2, we hoped to rescue the improved starvation survival of the swapped promoter strain by inducing higher expression of the AD-sector with the addition of cAMP to the pre-starvation medium. Indeed, the addition of cAMP to the pre-starvation medium improved survival (Fig 4E, 3rd bar from the top). However, this improved starvation survival came at the cost of a reduced growth rate on mannose minimal medium with cAMP (S4 Fig). According to our model, this slower growth rate is due to the protein expression burden of the increased AD-sector (Fig 2C).

Finally, we investigated cell motility, which is another adaptability phenotype that serves the purpose of escaping the current environment using different motility metrics on soft agar plates (Fig 5A). Motility also requires a substantial investment of proteomic resources in flagellar proteins [6]. The abundances of motility proteins increase with decreasing nutrient quality based on the dataset by Hui et al. [6], as shown in Fig 5B. Indeed, we found that wildtype bacteria growing on mannose were substantially more motile than wildtype bacteria growing on glucose as quantified by the increased fraction of colonies that showed large scale swarming (Fig 5C, blue circles vs. orange circles), as well as by an increase in colony range extension (Fig 4D, blue circles vs. orange circles. In contrast, the swapped promoter strain lost this increased motility when growing on mannose according to both metrics (Figs 5C and 4D, orange triangles). Like the wildtype strain on glucose, the swapped promoter strain on mannose formed small colonies with smooth edges (Fig 5A), and no colonies exhibited swarming (Fig 5C). However, when we artificially induced higher expression of AD-sector with addition of cAMP to the plates, the swapped promoter strain regained high motility (Figs 5C and 4D, inverted triangles). As demonstrated in batch culture, this higher motility comes at the cost of a reduced growth rate of the swapped promoter strain on mannose (S4 Fig). Hence, microbial motility closely matches the nutrient quality dependence of other adaptability phenotypes (Fig 4). We note that while Figs 3 and 4 show that gene expression patterns and phenotypes are consistent with our proposed hypothesis, they should not be considered as causal evidence for the slow growth rate on mannose.

## Discussion

Intriguingly, despite ecological complexity, global cellular resource allocation can be quantitatively described by surprisingly simple bacterial growth laws. We hypothesize that the simplicity of the growth laws is in fact a consequence of an elegant regulatory strategy, in which nutrient quality constitutes a convenient regulatory lever controlling resource allocation in specific ecological environments. As illustrated in Fig 6, substrates serve as signals that microbes use to identify the environment in which they find themselves, and substrate quality is the regulatory mediator of resource allocation strategies. Rather than reflecting fundamental biochemical or biophysical limitations like protein cost or limitations in capacity for transporters in the membrane, nutrient quality reflects a probabilistic map of the safety, reliability, and profitability of different ecological environments shaped by evolution. Low nutrient quality substrates thus indicate environments where the carbon source may quickly be depleted,

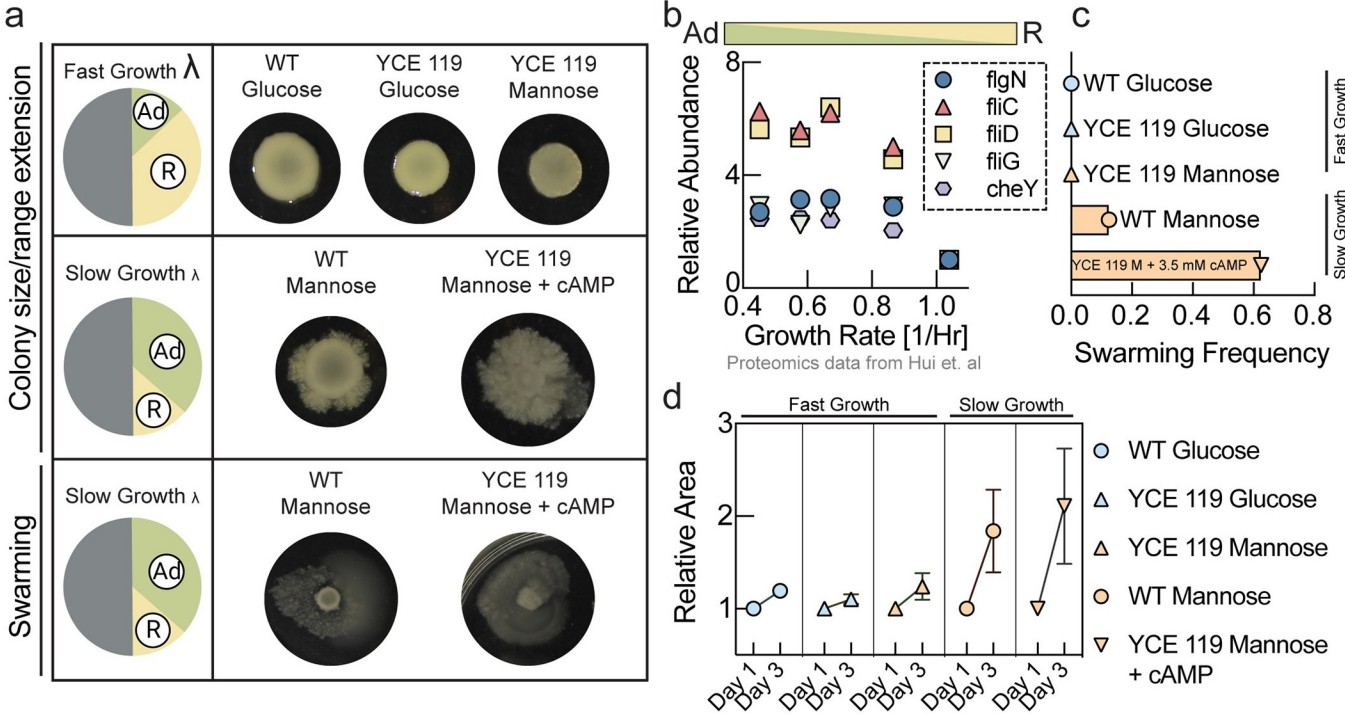

**Fig 5. Motility cost of higher nutrient quality. a,** Motility assays on soft agar plates. Colonies on fast growth medium are smaller and show smooth edges (top panel). Colonies on slow growth medium are larger with ruffled edges (middle panel). A fraction of colonies on slow growth medium shows a longer-range swarming phenotype (bottom panel). (pie chart illustrations created with Biorender.) **b,** Relative quantification of proteins related to motility. Proteomics data from Hui et al. [6]. (color bar illustration created with Biorender.) **c,** Fraction of colonies exhibiting swarming on soft agar. For wildtype on glucose (blue circle), we observed no swarming, whereas on mannose a substantial number of colonies showed swarming (orange circles). The swapped promoter strain (YCE 119) lost the ability to swarm on mannose (orange triangles), but addition of cAMP (3.5mM) resulted in a large fraction of swarming colonies. 24 colonies were analyzed for WT grown in Glucose, 72 colonies were analyzed for WT grown in Mannose, 24 colonies were analyzed for YCE 119 grown in Glucose, 48 colonies were analyzed for YCE119 grown in mannose, and 24 colonies were analyzed for YSE 119 grown in Mannose with 3.5mM cAMP. **d,** We tracked growth of colony size over 3 days on minimal medium soft agar plates. Colony sizes of the wildtype increased much more on mannose (orange circles) than on glucose (blue circles) and exhibited rough irregular edges indicative of higher motility. The swapped promoter strain lost this enhanced motility on mannose and only showed an increase in colony size similar to the wildtype on glucose (yellow triangles). However, with the addition of cAMP (3.5mM) to mannose medium, the swapped promoter recovered high motility (yellow inverted triangles).24 colonies were analyzed for WT grown in Glucose, 72 colonies were analyzed for WT grown in Mannose, 24 colonies were analyzed for YCE 119 grown in Glucose, 48 colonies were analyzed for YCE119 grown in Mannose, and 24 colonies were analyzed for YSE 119 grown in Mannose with 3.5mM cAMP. Normalized mean area fold change plotted with standard deviation).

making investment in alternative pathways worthwhile, or environments that tend to co-occur or precede other forms of stress that the bacterium should prepare for. While nutrient quality is correlated with the efficiency of transporters and metabolic enzymes reflected in protein investment, it is not fundamentally limited by these efficiencies in physiological conditions, at least for certain substrates. In this work, we demonstrated this for mannose, the 'poorest' substrate of *E. coli* that we know. We also collected prior evidence from the literature, demonstrating substantial growth rate improvements on the intermediate quality substrates, glycerol, and fructose, with resulting growth rates similar to those on glucose (S5 Fig). While these are isolated examples, and it is impossible to exhaustively demonstrate the plasticity of nutrient quality for all substrates of *E. coli*, these data suggest that sub-maximum growth on mannose is far from the exception and much more common than commonly appreciated. We propose here that sub-maximum growth rates are actually the norm and that growth rates in most growth conditions can likely be improved substantially.

We argue that nutrient quality constitutes a dial that can be quickly adapted by evolution in response to changing ecological conditions. This is consistent with experimental observations

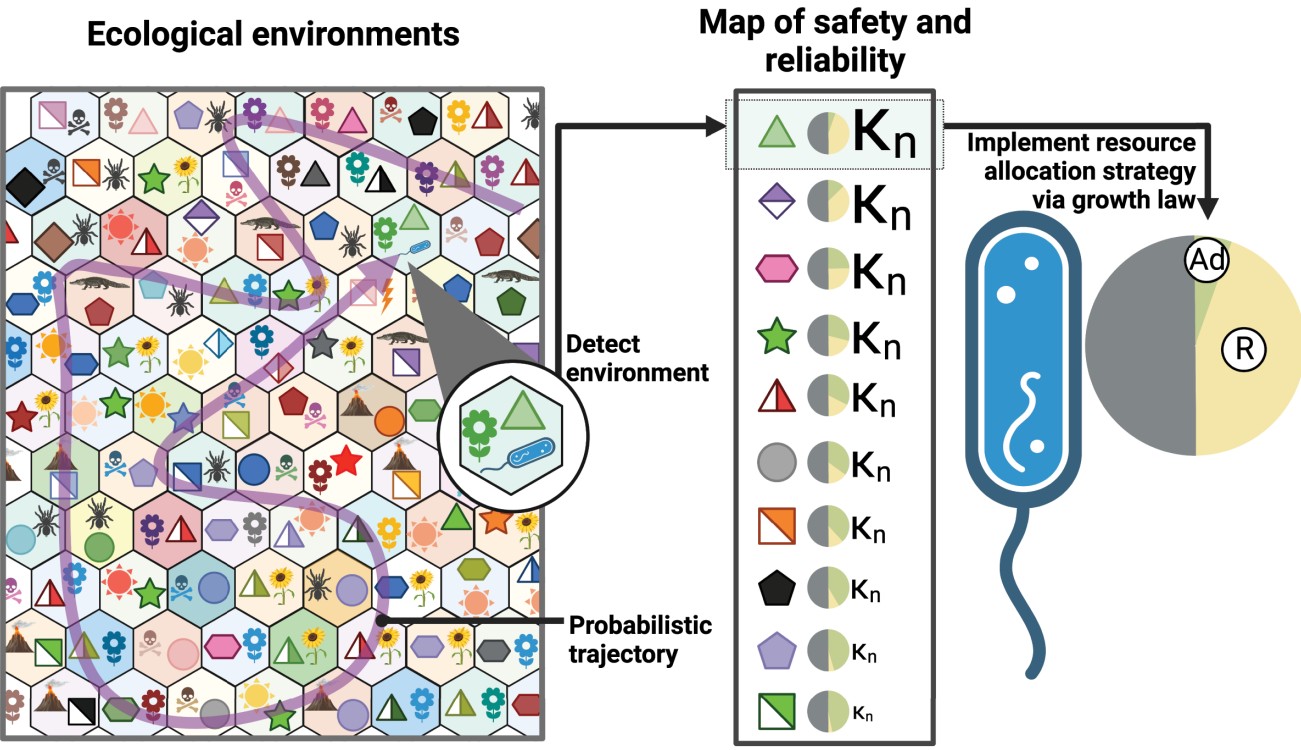

**Fig 6. Nutrient quality as a map of environments and mediator of resource allocation strategies.** Ecological niches in which bacteria like E. coli evolved consist of many strikingly different environments, each coming with their own set of payoffs and hazards. The trajectory in which the bacterium transitions between these environments is partly governed by its natural life cycle, but also has a probabilistic component. To maximize their fitness, bacteria need to identify the environment in which they find themselves and adjust proteome resource allocation strategies accordingly. Nutrients allow bacteria to grow, but they also serve as a major signal that allows microbes to infer information about their environment. Nutrient quality encoded in regulatory architecture and enzymatic properties that were shaped by evolution, serves both as a map of the safety and reliability of the environment and as a regulatory mechanism implementing proteome allocation decisions. The conservation of this simple regulatory architecture across environments is what gives rise to the striking bacterial growth laws and their elegant predictions. (Illustration created with Biorender).

that adaptive laboratory evolution can quickly lead to more rapid growth even on 'good' carbon sources like glucose [24] and that single point mutations can sometimes lead to substantially enhanced growth rates [25]. Crucially adaptation to changing environments by modifying nutrient quality does not disrupt finely tuned cellular programs coordinating different parts of metabolism. Thereby, this regulatory architecture allows bacteria to be highly adaptable to changing environments yet retain important optimized regulatory programs coordinating central metabolism, adaptability, stress response, and ribosomal flux. Such an integration of substrate-specific and global regulatory systems as a modulator of risk-dependent gene expression has been previously suggested [19]. Here, however, we argue that rather than being implemented as a consequence of 'energy'-rich or 'energy'-poor substrates, this strategy may be the central determinant of substrate quality itself.

## Supporting information

**S1 Fig. Protein cost of substrate-specific transporters and enzymes.** Protein copy number can be converted to protein cost in units of number of amino acids by multiplying copy number with the number of amino acids in each protein. A similar inverse correlation as the one

observed for copy number in Fig 1B of the main text, also holds for protein cost.
(DOCX)

**S2 Fig. Growth rate effect of individual genetic modifications and combinations.** Growth rates on glucose and mannose minimal medium of different strains. Data points are biological repeats. Unpaired t-test with welch correction was performed. (please see S3 Table for detailed description of strains).
(DOCX)

**S3 Fig. Crp activity at maximum growth rates for cAMP titrations.** In Fig 2E of the main text, we plotted endogenous CRP activity measured by Towbin et al. [17] against growth rate on the respective substrate. Endogenous CRP activity and CRP activity where growth rate is maximum are not always the same, as shown by Towbin et al. [17]. Therefore, we plot CRP activity at maximum growth rate against maximum growth rate from Towbin et al. [17] here and find a similar relationship.
(DOCX)

**S4 Fig. Growth rate of swapped promoter strain with the addition of cAMP.** As shown in the main text, many adaptability phenotypes that are lost in the swapped promoter strain (YCE 119) can be rescued by the addition of cAMP (3.5mM) to the growth medium, which upregulates the AD-sector. However, these improvements come at the cost of a reduced growth rate due to the substantial protein cost of the AD-sector.
(DOCX)

**S5 Fig. Growth rate improvements on other substrates. Left panel:** Point mutations in glycerol kinase glpK result in faster growth rates on glycerol [25–27]. Resulting growth rates are effectively identical to growth rates of the wildtype on glucose. Data was replotted from Basan et al. [7]. Note that these experiments were performed in MOPS buffered medium, that results in somewhat faster growth rates in all conditions. **Right panel:** Knockout of the regulator of gluconeogenesis Cra results in faster growth rates on glucose and mannose. Growth rate of the Cra knockout strain on glucose is effectively identical to the growth rate of the wildtype on glucose. Data was replotted from Basan et al. [22].
(DOCX)

**S1 Table. Protein copy number calculation.** Protein copy number calculation of the main carbon transporting enzyme (or the first enzyme in the primary carbon degradation pathway) from Li at. al. [12].
(DOCX)

**S2 Table. Protein copy number fold changes across growth rates.** Fold change calculation of the main carbon transporting enzyme (or the first enzyme in the primary carbon degradation pathway) at slow growth rate (0.45) from Hui. et. al. [6].
(DOCX)

**S3 Table. Detailed description of the strains used in S2 Fig.**
(DOCX)

## Author Contributions

**Conceptualization:** Avik Mukherjee, Markus Basan.

**Data curation:** Avik Mukherjee, Yanqing Huang, Markus Basan.

**Formal analysis:** Avik Mukherjee, Yanqing Huang.

**Funding acquisition:** Markus Basan.

**Investigation:** Avik Mukherjee, Yu-Fang Chang, Yanqing Huang, Nina Catherine Benites, Leander Ammar, Jade Ealy, Mark Polk, Markus Basan.

**Methodology:** Avik Mukherjee, Yu-Fang Chang, Markus Basan.

**Project administration:** Markus Basan.

**Resources:** Avik Mukherjee, Yu-Fang Chang, Markus Basan.

**Supervision:** Avik Mukherjee, Markus Basan.

**Visualization:** Avik Mukherjee.

**Writing – original draft:** Avik Mukherjee, Yu-Fang Chang, Yanqing Huang, Nina Catherine Benites, Jade Ealy, Mark Polk, Markus Basan.

**Writing – review & editing:** Avik Mukherjee, Yu-Fang Chang, Yanqing Huang, Nina Catherine Benites, Leander Ammar, Jade Ealy, Mark Polk, Markus Basan.

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
