## [Decision Letter · Decision Letter 0]

6 Oct 2023

Dear Dr. Basan,

Thank you very much for submitting your manuscript "Plasticity of growth laws tunes resource allocation strategies in bacteria" for consideration at PLOS Computational Biology.

As with all papers reviewed by the journal, your manuscript was reviewed by members of the editorial board and by several independent reviewers. In light of the reviews (below this email), we would like to invite the resubmission of a significantly-revised version that takes into account the reviewers' comments.

While the premise of this manuscript is very interesting, there are three very major concerns/issues with this manuscript, which the reviewers correctly point out.

1) Reviewer 1 (and 2) correctly point out that too many assumptions are made while using very limited carbon sources (only hexoses), and the experiments performed do not directly address the question (or allow alternate interpretations which are not considered).

2) With the model, there is very little/no model testing. Additionally, the model relies mostly on 'intuition'. This is great, but without substantial model testing and/or formalization it becomes difficult to extrapolate too much.

3) There is no direct genetic testing or evidence, nor a sampling of diverse carbon sources/conditions to see if there can be such strong universal statements made.

Separately, the study refers to a very small, selective body of work. Particularly, there have been some exceptional studies in the late 90-early 2000s, that better connect ribosomal expression with growth (and contextualize it in different growth conditions and/or bacteria).

We cannot make any decision about publication until we have seen the revised manuscript and your response to the reviewers' comments. Your revised manuscript is also likely to be sent to reviewers for further evaluation.

Sincerely,

Sunil Laxman, PhD

Academic Editor

PLOS Computational Biology

Pedro Mendes

Section Editor

PLOS Computational Biology

While the premise of this manuscript is very interesting, there are three very major concerns/issues with this manuscript, which the reviewers correctly point out.

1) Reviewer 1 (and 2) correctly point out that too many assumptions are made while using very limited carbon sources (only hexoses), and the experiments performed do not directly address the question (or allow alternate interpretations which are not considered).

2) With the model, there is very little/no model testing. Additionally, the model relies mostly on 'intuition'. This is great, but without substantial model testing and/or formalization it becomes difficult to extrapolate too much.

3) There is no direct genetic testing or evidence, nor a sampling of diverse carbon sources/conditions to see if there can be such strong universal statements made.

Separately, the study refers to a very small, selective body of work. Particularly, there have been some exceptional studies in the late 90-early 2000s, that better connect ribosomal expression with growth (and contextualize it in different growth conditions and/or bacteria).

Reviewer's Responses to Questions

**Comments to the Authors:**

Reviewer #1: In the paper "Plasticity of growth laws tunes resource allocation strategies in bacteria" the authors show that differences in growth of bacteria between nutrient sources of different "quality" are not a result of the nutrient's inherent metabolic property but likely a result of sub-optimal gene expression resource partitioning. They demonstrate this in particular for mannose as a carbon source. Mannose being a hexose that converts to fructose-6-phosphate thus entering glycolysis, their ultimate conclusion is fairly intuitive, but it is good to have it demonstrated. I have the following comments on the work.

1. The authors should interpret their findings in light of this classic work from Fred Blattner's lab, published nearly 20 years ago. https://www.jbc.org/article/S0021-9258(20)69269-9/fulltext. This work shows how the transcriptional program of E. coli progressively activates more genes as the 'quality' of carbon source declines. It also shows how RNAP clustering at the highly expressed rRNA genes also declines as the c-source quality decreases. Both these findings are intimately related to the present work. In addition, Blattner's team's work has also used a diverse set of carbon sources.

2. I feel that the general conclusion that the authors have come up with will need more support from different categories of carbon sources, similar to the above-cited work from Blattner's laboratory. If not, it should be clearly stated in the abstract and introduction that their experimental proof applies to hexoses and not broadly among different substrates / nutrients - this itself is valuable.

3. The authors discuss protein cost purely from its expression levels. However, I understand that the bacteria were grown in minimal media without external amino acids. In this case, there should also be cost arising from the biosynthesis of amino acids. This, to the best of my understanding, has not been incorporated.

4. The authors activate mannose metabolism using three different mutations, one of which - introducing a ter promoter- appears to be not the most relevant to sugar metabolism. It appears that Mlc, which the authors have deleted, regulates some PTS components involved in mannose metabolism as well. These make their results difficult to interpret. What are the effects of the individual mutations?

5. An alternative approach would be to perform laboratory evolution of the bacterium in mannose as the sole carbon source. I am pretty sure it would find mutations that make it grow better. This in itself would indicate that slow growth in mannose is not because of mannose being mannose. Characterisation of individual mutations arising and their combinations thereof would be a powerful approach towards characterising various strategies that enable the bacterium to deal with the bottlenecks that stultify growth in mannose.

Reviewer #2: In this contribution the authors explore the notion of "nutrient quality", as it appears in resource allocation models, based on experimental results and analysis of data from the literature. The main question is to assess whether "nutrient quality" is determined by fundamental biochemical limitations or rather by a genetic program shaped by evolution and potentially adaptable to new environments. By analyzing the response of a mutant with upregulated growth on a poor carbon source, and its adaptability phenotypes, they conclude that "nutrient quality" is not an intrinsic biochemical property but rather the result of evolution.

I find the study interesting, but with main limitations as follows:

- The three-compartment resource allocation model that authors start from has been established based on a linear constraint among the size of the three compartments that manifests itself from experimental evidence. In the authors contribution, this model is refined into a multiple compartment model which by nature also corresponds to introducing linear relationships among the compartments. However, the authors' arguments seem to me mostly qualitative, not quantitative. Thus the refined model seems a bit vague to me: How does the size of the many compartments change in different conditions, and how do these changes relate with growth rate?

This concern also relates with the next.

- For the targeted venue, what is the computational element (or contribution) of the paper? It seems to me that, besides standard data analysis, the computational nature is limited to the review of an existing growth rate /resource allocation model that is later discussed and refined purely based on qualitative considerations

- Authors base their experimental evidence on the construction of a mutant with faster growth on mannose, which amounts to replacing the chromosomal promoter driving a mannose PTS permease with the promoter of the glucose PTS transporter PptsG. This very circumscribed genetic modification seems to me in contrast with the authors' statement at the end of the introduction that "[changing] growth rates by simply changing the expression level of carbon-specific transporters and metabolic enzymes [is] challenging [...] because metabolic pathways of individual substrates typically involve many enzymes and transporter that need to be expressed in concert". Turned another way, how robust is the experimental analysis to the fact that changes in nutrient quality cannot be confined to changes in a single element of the corresponding metabolic chain?

- I find the term "nutrient quality" confusing for the discussion that the authors are making. For instance, "nutritional capacity" (as per your reference [4]) seems better adapted to me, since this expression is not solely associated with the nutrient itself (as in "nutrient quality") but rather with the nutrient and the nutritional efficiency resulting from the corresponding metabolism. Also I wonder why the authors do not explicitly discuss the chemical composition of substrates as a possible defining part of their "nutritional capacity"; Is this aspect implicitly captured by the hypothesis that "Nutrient quality could be an intrinsic carbon-specific property such as proteome cost of the catabolic pathways, required for metabolizing the substrate and therefore determined by

fundamental enzymatic properties like the maximum catalytic rate V max of these metabolic enzymes and transporters."? This hypothesis seems more related with the cost of metabolizing the substrate rather than with the intrinsic substrate richness

- Partly but not only because of the previous issue, I find the first part of Results unclear. For instance, in the context where this is discussed, it is quite unclear to me why authors say that the rank-ordered anticorrelation between the copy number of carbon transporters and growth rate is a striking finding (¨"Strikingly, when we plotted..."). I find that results and their interpretation starts being clear when starting to discuss results on the swapped promoter strain (from "Next, we measured growth rates of swapped promoter strain..." )

- Statement "We propose that different substrates convey information about the safety and reliability of the current growth environment", and related statements: Why "safety and reliability"? I rather understand your arguments in terms of evolution having shaped the genetic program so as to avoid focusing cellular resources on substrates that are on average of little interest, while saving cellular resources to adapt fast to substrates that are on average more profitable

- Fig.3: Wouldn't it be interesting to also report how YCE behaves on glucose? Are the results in such scenario in line with your expectations?

- I find the title generic and unclear about the point the authors make. I also find the Author summary too similar to the Abstract, it should be more sharply different (less technical) in spirit

Besides these main issues, I find the paper reasonably well written, except for a few minor typos / lexical inconsistencies scattered in the text that I invite authors to fix.

Reviewer #3: I performed this review with the help of a junior group member.

This study by Mukherjee and coworkers uses E. coli experiments and literature data to challenge the concept that fundamental biochemical constraints regulate nutrient quality, the catabolic flux per enzyme mass measured for growth in different defined media. The central finding is that E. coli can achieve comparable growth rates on mannose and glucose when the regulatory system controlling mannose transport is swapped with that of glucose. Consequently, it does not seem appropriate to view mannose as a poorer growth medium.

Instead, the authors conclude that the concept of nutrient quality should be understood as a regulated and ecological property, influenced by the organism's adaptation and preference. In other words, nutrient quality signifies how well a substrate aligns with the metabolic capabilities and regulatory mechanisms of bacteria within a specific ecological context.

I find the study interesting and definitely worth publication on Plos CB, but I have a number of comments that the authors may want to address - listed below.

I did not find the order in which things are presented very clear. Most of all it seems important to establish clear distinctions between novel findings and novel hypotheses introduced in this study, and existing findings and hypotheses from prior literature. For example, in Box 1 the distinction between new hypotheses and ingredients taken from the literature is not very clear.

Experiments and data analysis from other datasets (mainly Hui et al) form the core of the work. The model, on the other hand, serves mainly as a conceptual framework, as the model itself and its predictions are not really fully formalized, and there is no clear comparison between the model's predictions and the experimental findings. This is ok I think but some of the claims do not appear commensurate with this fact. Adding some clarifications could help the reader.

Since no real model testing is performed, it would be seem more accurate to characterize Figures 3 and 4 as consistent with a hypothesis rather than as conclusive evidence towards a model prediction. Protein functional classes that associate to lag, starvation, motility, show trends (Fig 3bc, 4b) in line with the authors' hypothesis, and so do the genetically engineered strains studied here (Fig 3de, 4cd)

Adding some formalization may also help. For example manuscript just gives some intuition on what k_n should be given the data. However, it misses a clear definition of "nutrient quality". How should an equation for k_n look like, given the presented results?

More conceptually, the model (if I understood correctly) suggests that the parameter kn, representing nutrient quality, can be adjusted through a specialized sector with minimal associated costs (which can be neglected). Additionally there exists a trade-off between increasing nutrient quality and a separate "adaptability sector" encompassing factors like motility and costs related to lag phases and starvation. A question arises regarding why this trade-off manifests differently for various substrates (glucose vs mannose). Additionally, it raises the question of why the wild-type bacteria can afford such a higher vulnerability by shutting down their adaptability sector in glucose-rich environments.

Fig 1 and S1 and their interpretation need to be better explained in the text. Currently it appears that some logic steps are missing. First, the fact that the substrates may or may not be present in the medium only appears very late in the text, after all the data have been presented (and it is not said for which data points this is the case, if not all). Second, the possible interpretation in terms of protein cost is dismissed almost before it is introduced and explained clearly. It seems this part would profit from a step-by-step narrative. Also sentences (plots?) detailing the expectations from protein cost are needed.

Specifically, the sentence "However, the observed anticorrelation is still surprising as these substrates were not present in the experimental growth conditions from which protein abundances were determined." is not clear. Do the authors use an extrapolation of the data in order to cover the new substrates? Why one would expect the data to not follow the growth law for the P sector? Can the authors show the comparison between their data in figure 1B and the growth law? Moreover, are the authors talking about the P sector as a whole or about the specific transporters and pathways of each considered substrate?

The explanation of the genetic system is also not very clear, and lacks a clear question statement e.g. "we ask whether growth rates of E. coli on mannose were limited by the cost of producing the necessary proteins for substrate transport and metabolism, or if they were limited by the available capacity for substrate transport". The PTS system is not defined.

Regarding Figure 3, it would be interesting to see if the relation between growth and death rate found in Biselli et al still holds for the "swapped" strain (the starvation procedure appears to be the same as in Biselli et al.)

Minor Remarks

In the paragraph after Figure 2, there is a bit of confusion regarding the sectors P, C and C*

**Have the authors made all data and (if applicable) computational code underlying the findings in their manuscript fully available?**

Reviewer #1: **No: **The authors haven't yet done this, but state that code will be uploaded to a suitable repository.

Reviewer #2: **No: **In Data and Code Availability, the authors declare: "All code will be uploaded to a public repository." and "Data underlying all figures will be uploaded as supplementary files or to a public repository." Unless I am mistaken, these uploads are not available yet (not linked from the submission).

Reviewer #3: **No: **They only state that they will be made available

PLOS authors have the option to publish the peer review history of their article (what does this mean?). If published, this will include your full peer review and any attached files.

Reviewer #1: No

Reviewer #2: No

Reviewer #3: **Yes: **Marco Cosentino Lagomarsino
---

## [Decision Letter · Decision Letter 1]

4 Dec 2023

Dear Dr. Basan,

We are pleased to inform you that your manuscript 'Plasticity of growth laws tunes resource allocation strategies in bacteria' has been provisionally accepted for publication in PLOS Computational Biology.

Best regards,

Sunil Laxman, PhD

Academic Editor

PLOS Computational Biology

Pedro Mendes

Section Editor

PLOS Computational Biology

Reviewer's Responses to Questions

**Comments to the Authors:**

Reviewer #1: The authors have made very reasonable efforts towards addressing my concerns by presenting additional data and/or by rewording their paper to better reflect their data. The paper is very nice.

Reviewer #2: This reviewer is satisfied with the response of the authors to the first review round, and with the improvements that have been made to the paper.

**Have the authors made all data and (if applicable) computational code underlying the findings in their manuscript fully available?**

Reviewer #1: Yes

Reviewer #2: Yes

PLOS authors have the option to publish the peer review history of their article (what does this mean?). If published, this will include your full peer review and any attached files.

Reviewer #1: No

Reviewer #2: No

---

## [Editor Report · Acceptance letter]

1 Jan 2024

PCOMPBIOL-D-23-01241R1 

Plasticity of growth laws tunes resource allocation strategies in bacteria

Dear Dr Basan,

I am pleased to inform you that your manuscript has been formally accepted for publication in PLOS Computational Biology. Your manuscript is now with our production department and you will be notified of the publication date in due course.

With kind regards,

Dorothy Lannert
